# Molecular Classification of Endometrial Cancer and the 2023 FIGO Staging: Exploring the Challenges and Opportunities for Pathologists

**DOI:** 10.3390/cancers15164101

**Published:** 2023-08-15

**Authors:** Wenxin Zheng

**Affiliations:** Department of Pathology, Obstetrics and Gynecology, University of Texas Southwestern Medical Center, Dallas, TX 75390, USA; wenxin.zheng@utsouthwestern.edu; Tel.: +1-214-648-1190

**Keywords:** endometrial cancer, FIGO staging, molecular classification, TCGA-EC, POLE mutations

## Abstract

**Simple Summary:**

In this commentary, I delve into the complexities and potential of the recently proposed 2023 FIGO staging system for endometrial cancer, with a focus on the incorporation of molecular classifications. I aim to evaluate the predictive capacity of histology and molecular testing, emphasizing the challenge of utilizing the POLE mutation test in determining prognosis. By addressing challenges like discerning synchronous independent tumors from metastatic ones when both endometrium and ovary are involved, and considering new elements such as lymph node metastasis size, this work may inform future treatment approaches, reinforcing the indispensable role of pathologists in this evolving landscape.

**Abstract:**

This commentary explores the complexities of the FIGO 2023 staging system and the inclusion of The Cancer Genome Atlas’s (TCGA) molecular classification in the management of endometrial cancer. It highlights the importance of histology as a prognostic tool, while scrutinizing the merits and demerits of its application to aggressive endometrial cancers. The commentary review sheds light on the recent introductions of lymphovascular space invasion (LVSI) and lymph node metastasis size in cancer staging. It outlines the difficulties in differentiating between synchronous and metastatic endometrial and ovarian cancers, underlining their implications on treatment strategies. Furthermore, the commentary discusses the integration of molecular classifications within the FIGO 2023 framework, emphasizing the pivotal yet challenging implementation of the pathogenic POLE mutation test. The commentary concludes by reaffirming the vital role of pathologists in executing the FIGO 2023 staging system.

## 1. Introduction

Endometrial cancer, the foremost gynecologic cancer in the United States, is currently the fourth most common cancer among women [1,2]. It is predicted that there were 65,950 new endometrial cancer cases in 2022, leading to an estimated 12,550 deaths [2]. This rising trend is not confined to high-income populations but has a significant impact on women globally, particularly those in lower-income brackets [2]. Studies project that by 2030, endometrial cancer will surpass colorectal cancer, making it the third most common and the fourth leading cause of cancer-related deaths among women [3]. This looming scenario emphasizes the global health burden posed by endometrial cancer and underscores the urgent need for effective interventions.

A significant advancement in diagnosing and treating endometrial cancers over the past decade has been the development of molecular classification systems. Molecular features can offer insights into the risk of recurrence and consequently survival outcomes [4,5,6]. Among such systems, The Cancer Genome Atlas (TCGA) provides a comprehensive classification, categorizing endometrial cancers into four distinct genomic categories: POLE ultramutated (POLE*mut*), microsatellite instability hypermutated or mismatch repair deficiency (MSI-H or MMRd), somatic copy-number alteration low or non-specific molecular profiler (CNL or NSMP), and the somatic copy-number high or p53 abnormal (CNH or p53*abn*) [4]. Each category corresponds to a different prognosis, with POLE*mut* cases typically exhibiting the best outcomes and p53*abn* the worst, while the remaining two categories present intermediate prognoses.

Recently, the FIGO 2023 endometrial cancer staging system has integrated this molecular classification [7], signifying a substantial step towards enhancing our ability to stratify endometrial cancer risks beyond the limits of traditional histologic evaluations. This new development offers a set of guidelines to aid clinicians in determining patient treatment options. However, the multiple pathologic parameter changes and the inclusion of molecular classifications in the FIGO 2023 update pose significant challenges, especially for pathologists. Despite these challenges, the evolving landscape also offers unique opportunities. Pathologists may find an increasingly important role in the diagnosis and management of endometrial cancer patients, provided institutions invest in necessary infrastructure and training to adapt to these advancements.

This commentary review seeks to illuminate the critical diagnostic aspects of the FIGO 2023 staging system and the role of TCGA’s endometrial cancer molecular classification in patient management. The objective is to enhance pathologists’ comprehension of these pivotal advancements, thus promoting effective communication and collaboration with clinicians during this transitional period.

### 1.1. Endometrial Cancer Staging in FIGO System

The International Federation of Gynaecology and Obstetrics (FIGO) staging system has been a critical tool in determining treatment decisions and predicting prognosis for endometrial cancer. Initially introduced in 1988 as a purely clinical system, it underwent a significant revision in 2009 to incorporate clinicopathological factors indicative of local, regional, and distant tumor spread [8]. Surgical staging typically includes total hysterectomy, bilateral salpingo-oophorectomy (THBSO), along with pelvic and para-aortic lymph node dissection. Additional procedures such as omentectomy, peritoneal biopsies, and peritoneal washings are generally reserved for non-endometrioid endometrial cancer (NEEC) patients. Accurate staging of individual endometrial cancer cases heavily relies on thorough pathological evaluations since identifying hidden microscopic metastases not only impacts staging but also provides crucial prognostic information and informs subsequent treatment strategies. Despite its broad application, the FIGO 2009 system unveiled several significant limitations over its decade of usage. These include insufficient attention to the histologic type of endometrial cancer, neglect of lymphovascular space invasion (LVSI), lack of differentiation based on nodal metastasis size, ambiguous staging assignments for patients with endometrioid carcinomas involving both the endometrium and ovary, and omission of molecular classification. The updated FIGO 2023 staging system has been developed to rectify these limitations and further refine the staging of endometrial cancer. By addressing these shortcomings, it aims to enhance patient care significantly.

### 1.2. Histology of Endometrial Cancer: An Independent Prognostic Marker

Endometrial cancers fall into two main histological categories: endometrial endometrioid carcinomas (EECs) and non-endometrioid carcinomas (NEECs). EEC is the most common subtype, accounting for approximately 75–80% of all endometrial cancer cases. It is typically associated with excessive estrogen exposure and generally has a favorable prognosis. Conversely, NEEC, which includes subtypes such as endometrial serous carcinoma (ESC), clear cell carcinoma (ECCC), carcinosarcoma, gastrointestinal carcinoma, squamous cell carcinoma, dedifferentiated carcinoma, undifferentiated carcinoma, mesonephric-like adenocarcinoma (MLA), and mixed endometrioid with serous or clear cell carcinomas, accounts for about 20–25% of all endometrial cancers. NEEC is not typically associated with excessive estrogen stimulation, and these cases often have a poor prognosis and exhibit more aggressive behavior [9,10]. Despite the impact of histologic type on prognosis, it was not incorporated into the 2009 FIGO staging system [8]. Recognizing its prognostic importance, the FIGO 2023 revision has integrated histologic type into the staging system, leading to different classifications and management strategies for EECs and NEECs. For instance, EECs and NEECs with similar levels of myometrial invasion are now staged differently: EECs as stage I and NEECs as stage II. This update underscores the increased need for pathologists to deliver more precise diagnoses of cancers’ histologic types.

High-grade endometrial cancers, specifically the differentiation between high-grade EEC versus ESC, often present a diagnostic challenge for pathologists [9]. For example, when three established gynecologic pathologists reviewed 56 high-grade endometrial cancer cases, only 64% of diagnoses were unanimous, while 36% diverged [11]. This discrepancy highlights the inherent limitations of microscopic diagnosis. To address this, the TCGA endometrial cancer molecular classification successfully distinguishes between these types by sorting them into the four categories: POLE*mut*, MMRd, NSMP, and p53*abn* endometrial cancers [12]. This molecular approach makes risk stratification straightforward and is one of the reasons molecular classifications have been incorporated into the FIGO 2023 staging system.

The 2023 FIGO staging system distinguishes endometrial cancers into two principal categories based on their degree of aggressiveness: aggressive and non-aggressive histologic types. The non-aggressive category largely encompasses low-grade or FIGO grade 1–2 endometrioid endometrial carcinomas (EECs), while the aggressive category includes all high-grade endometrial cancers, encompassing FIGO grade 3 EEC (G3 EEC) and all NEECs [7]. However, it is important to note that not all G3 EECs demonstrate aggressive characteristics. These tumors have also been classified into the four categories established by TCGA molecular classification [12]. In an influential study, Bosse T. with his colleagues and collaborators examined 381 G3 EEC cases, identifying the following subtypes: 49 (12.9%) POLE*mut*, 79 (20.7%) p53*abn*, 115 (30.2%) NSMP, and 138 (36.2%) MMRd tumors [12]. Their research suggests that only about 20% of grade 3 EECs with the p53*abn* subtype can genuinely be labeled as aggressive. In contrast, the remaining 80% of G3 EECs may be subject to overstaging and/or overtreatment when molecular classification is not implemented. This circumstance also extends to other aggressive endometrial cancers including ECCCs [13,14,15], undifferentiated or dedifferentiated carcinomas [16], and even carcinosarcomas [17]. Despite their typically aggressive behavior, these cancers also consist of four molecular subtypes when classified using the TCGA molecular approach [18,19]. It is worth noting, however, that the number of POLE*mut* cases is significantly fewer in these categories than in grade 3 EECs. The findings highlight the crucial importance of molecular classifications, particularly as the FIGO 2023 staging is being actively applied in clinical settings.

This discussion aligns with the recommendation in the FIGO 2023 statement, which encourages a comprehensive molecular classification for all cases of endometrial carcinoma. This approach is designed to aid in stratifying prognostic risk groups and guide treatment decisions. In line with this, it is advocated that molecular subtype evaluation should be incorporated into the staging criteria when feasible, as it can lead to a more accurate prognosis prediction. However, the stark reality is that molecular classification is not widely available in most institutions or hospital laboratories worldwide. As a result, pathologists often find themselves needing to classify cancer histologic types as precisely as possible with the resources available. In the sections that follow, experiences and approaches are shared for diagnosing and differentiating some clinically significant entities that frequently present diagnostic challenges.

### 1.3. Endometrial Serous Carcinoma

Endometrial serous carcinoma (ESC) is often viewed as a prototype for aggressive endometrial cancer, as the majority of ESCs, marked by p53 mutations and lacking POLE ultramutations, fall into the p53*abn* category. However, it is vital to understand that molecular classification does not directly assist in the diagnosis of ESC. Instead, it aids in the identification of serous-like or serousoid [9] endometrial cancer. Moreover, it is crucial to comprehend that serousoid endometrial cancer, as discerned through molecular classification, does not equate to ESC and includes a significant proportion of p53-mutant EECs [9]. The diagnosis of ESC is strictly made through microscopic or morphologic examination (Figure 1).

ESC typically manifests through papillary, glandular, and solid growth patterns with high nuclear grade and prominent nucleoli. About 70% of ESCs exhibit the classic serous morphology, which can be identified based purely on the morphological characteristics. Biomarkers, though helpful in pathology practice, do not definitively establish an ESC diagnosis. It is important to reiterate that mutant p53 expression patterns do not equate to an ESC diagnosis, as numerous endometrial cancers may display a mutant p53 immunophenotype. Endometrial carcinomas with ambiguous morphology, accounting for roughly 30% of cases and characterized as tumors displaying partial attributes of both grade 3 (G3) EEC and ESC [20] (Figure 2), often undergo a panel of immunohistochemistry (IHC) tests. These typically include PTEN, p53, p16, and mismatch repair (MMR) tests, especially when POLE mutation status is unknown. If the results reveal intact PTEN, proficient MMR, mutant p53, and block-pattern p16, a diagnosis of serousoid carcinoma is favored. Alternatively, any deviations from these IHC results tend towards a G3 EEC diagnosis. The detection of a POLE mutation status also supports a G3 EEC diagnosis in this setting.

### 1.4. Endometrial Clear Cell Carcinoma

Endometrial clear cell carcinoma (ECCC) constitutes another aggressive subtype of endometrial cancer. It presents distinct morphological patterns, including tubulocystic, papillary, and solid growth within a hyalinized stroma. These unique morphological features typically facilitate a straightforward microscopic diagnosis. Nonetheless, in instances of diagnostic ambiguity, the use of biomarkers such as napsin A, AMACR, and HNF1B can assist in confirming the diagnosis [9].

Endometrioid endometrial carcinomas (EECs) do not consistently display aggressive behavior, emphasizing the importance of understanding their heterogeneity. Studies have shown that these cancers can be categorized into the same four molecular subtypes identified in other high-grade endometrial cancers: POLE*mut*, MMRd, NSMP, p53*abn* [13,14,15,18,19]. It is noteworthy that about one-third of the studied EECs fall into the p53*abn* category (Figure 3), and up to 6% are classified as POLE*mut* cases [13,14,15]. Significantly, all ECCC cases with POLE mutations have shown prognoses similar to other POLE*mut* endometrial cancers. To prevent potential overestimation of ECCC severity, molecular classification is recommended whenever feasible.

However, an important distinction should be made: ECCCs with wild-type p53 in the NSMP group do not have the same behavior as NSMP wild-type p53 EEC cases. The prognosis for NSMP ECCCs is notably worse, underscoring the need for detailed morphological analysis even when molecular classification is employed [15].

### 1.5. Endometrial Carcinosarcoma

Endometrial/uterine carcinosarcoma, an aggressive high-grade endometrial cancer, has a biphasic nature, encompassing a sarcoma element that has dedifferentiated from the carcinoma element. Although relatively rare, the incidence has been on an upward trend, accounting for about 5% of all endometrial cancers [21]. It is usually detected in elderly individuals, although an uptick in younger patients has been observed in recent years. The stage of the tumor has shifted, with nodal metastases becoming more common and distant metastases decreasing (http://seer.cancer.gov/ (accessed on 29 July 2023)).

Genomic studies have started to unravel the mutational profile of carcinosarcoma, which has been less studied compared to general endometrial cancers. Both the carcinomatous and sarcomatous elements seem to share concordant genetic mutational profiles. Recent next-generation sequencing studies have revealed that carcinosarcomas can be serous and endometroid, poorly differentiated, and bear common somatic mutations in genes such as TP53, PIK3CA, FBXW7, PTEN, and ARID1A [22]. The mutation rates of these genes fluctuate based on the grade of the carcinomatous components [22].

The histological element of carcinosarcoma, primarily high-grade, significantly contributes to disease progression and metastasis. Less commonly, the carcinoma component can be low-grade, constituting approximately 20% of all carcinosarcomas (Figure 4). Sarcoma dominance, a recently recognized concept defined by a sarcomatous element comprising more than 50% of the tumor, is seen in 40% of carcinosarcoma cases. Similarly, a heterologous element is also present in 40% of carcinosarcomas, and both conditions are associated with a poorer survival prognosis. The sarcoma component, thought to be dedifferentiated from the carcinoma component, impacts local tumor spread, such as cervical stroma, vaginal, or adnexal involvement, while the carcinoma component is primarily responsible for most nodal and distant metastases. The epithelial–mesenchymal transition (EMT) plays a crucial role in the sarcomatous dedifferentiation in carcinosarcoma, and heterologous sarcoma displays a higher EMT signature than its homologous counterpart.

Originally, carcinosarcoma was not included in the TCGA molecular classification. However, a recent meta-analysis of 263 uterine carcinosarcomas across five studies revealed that POLE*mut* carcinosarcomas have a prognosis as excellent as that of POLE*mut* EECs [17], with all POLE*mut* carcinosarcomas containing non-serous carcinomatous components. This finding underscores the significance of implementing molecular classification for carcinosarcomas, particularly those with non-serous carcinomatous components. From a standard pathology practice perspective, I advocate that reporting of carcinosarcoma should encompass the histologic types and proportions of both the carcinomatous and sarcomatous elements.

### 1.6. Endometrial Mesonephric-like Adenocarcinoma

Endometrial mesonephric-like adenocarcinoma (MLA) is a relatively new and uncommon subtype, making up around 1% of all endometrial cancer cases [23,24]. This often-overlooked endometrial cancer subtype presents considerable challenges when it comes to diagnosis. Since the recognition of MLA occurred post-TCGA molecular classification, it was not included in the TCGA endometrial cancer study [4]. However, due to its inherently aggressive nature, MLA has recently been acknowledged as an additional aggressive histological type of endometrial cancer in the FIGO 2023 system [7].

The exact cellular origin of MLA remains a topic of active discussion. Morphological resemblances and similar immunohistochemical characteristics to classic mesonephric carcinoma might lead some to consider it as a variant of mesonephric carcinoma with divergent Müllerian attributes. However, the frequent co-existence of MLA with Müllerian anomalies, such as endometrial hyperplasia and adenomyosis, suggests a Müllerian origin.

Morphologically, MLAs exhibit a broad array of growth patterns, from small tubules and ductal/glandular formations to various other structures within and across tumors [25]. As a result, they can be erroneously identified and diagnosed as low-grade EEC, ECCC, ESC, or even carcinosarcoma. Although inconsistent, luminal eosinophilic colloid-like secretions are often discernible. On the cytological front, MLAs typically present mild to moderate atypia, with occasional high-grade atypia. Immunohistochemically, MLAs are usually found to express markers such as PAX8, GATA3, TTF1, calretinin, and CD10. On the other hand, they typically do not express ER/PR, are proficient in MMR, and carry wild-type p53 [24] (Figure 5). Among these markers, GATA3 and ER/PR have been identified as the most useful diagnostic biomarkers for MLA [26]. In standard pathology practice, suspicion of MLA often arises when an endometrioid-looking carcinoma exhibits negative ER/PR staining. Current data suggest that MLA would most likely be classified in the NSMP category as it is typically p53 wild-type, MMR proficient, and not part of the POLE*mut* endometrial cancer group due to its high recurrence rate and frequent distant metastasis. It is worth noting that ER/PR negative EECs have been shown to have a poorer prognosis than their grade- and stage-matched ER/PR-positive counterparts [27]. Given that MLAs often lack ER/PR expression, this characteristic could be associated with their notoriously poor prognosis.

For MLAs, I do not advocate for molecular classification once the pathologic diagnosis has been confirmed. This is the only aggressive subtype of endometrial cancer where I take this stance. Advances in understanding MLA’s pathogenesis and molecular classification are anticipated to improve our ability to manage this unique subtype of endometrial cancer.

## 2. Recent Advancements and Insights in Endometrial Cancer with Lymphovascular Space Invasion

Lymphovascular space invasion (LVSI), a factor that pathologists frequently assess in hysterectomy specimens, has recently garnered increased attention in the risk stratification systems for endometrial cancer [28,29]. Previously, the presence of LVSI, irrespective of the depth of myometrial invasion, was a criterion recommending adjuvant radiation treatment for patients with stage I grade 1 or 2 EEC [29]. The FIGO 2023 staging system [7], although not previously necessary for staging, now includes LVSI as a determinant for stage II endometrial cancers, particularly when significant LVSI is found in what are otherwise classified as stage I diseases [30,31].

This alteration in the FIGO 2023 staging system emphasizes the enhanced prognostic significance that substantial or extensive LVSI holds over focal or no LVSI. Consequently, precise documentation of the extent of LVSI by pathologists becomes imperative. This need is primarily driven by the absence of quantifiable diagnostic criteria for LVSI and differing definitions and assessment approaches found in the literature. A few essential points for deciphering LVSI’s morphological mimics are outlined here. The most frequent LVSI mimic is the artefactual displacement of tumor fragments within myometrial clefts or large-caliber endothelial-lined vessels, often due to uterine manipulation during surgery [32]. This displacement is likely to occur in cases with poor fixation or endometrial cancers exhibiting tumor necrosis. Appearances that may also imitate LVSI include stromal retraction around invading tumor cells, degenerative tumor cells with inflammatory cells inside lymphovascular spaces, clusters of tumor cells resembling the primary tumor without altered tumor cytology, and superficial locations just beneath the endometrial cancers without clear myometrial invasion.

Distinguishing true LVSI from its mimics can be challenging, with several histological criteria suggested, such as proximity to a large venous and arterial vessel, presence of perivascular lymphocytes, and altered tumor cytology compared to the main tumor morphologic features. In our routine practice, we view definitive tumor cells within an endothelial-lined channel, absent any features suggesting the above-mentioned artefactual vascular invasion, as indicative of genuine LVSI.

The FIGO 2023 guidelines emphasize the importance of evaluating the extent of LVSI, as focal or no LVSI correlates with better prognosis, while substantial LVSI is associated with poorer prognostic outcomes. “Substantial LVSI” has been variably defined, including identifying four versus five or more instances of LVSI on a single H&E glass slide, located at the tumor periphery and not within the tumor mass itself [7,33]. Following the WHO 2020 guidelines [34], the FIGO 2023 system recommends five or more instances of LVSI as the criterion for substantial LVSI. This criterion from FIGO 2023 should be utilized when evaluating endometrial cancer cases.

Determining the count of LVSI instances is usually achievable by following the aforementioned microscopic criteria. However, the rigidity of the criteria for substantial LVSI could pose an issue when the number of LVSI instances falls just below 5, for instance, 3–4 counts on a single slide. At present, there is a paucity of data on how these sub-criteria conditions correlate with prognosis and survival. Therefore, it is advisable to document such findings in pathology reports for future analysis and research and to discuss the findings with oncologists in tumor board meetings.

## 3. The Importance of Lymph Node Metastasis Size in Endometrial Cancer

Surgical staging of endometrial cancer classifies cases with nodal involvement as stage IIIC, indicating the need for adjuvant therapy to minimize risks of recurrence and metastatic disease. Despite the diagnostic value of lymphadenectomy, its routine use in managing endometrial cancer has been a subject of debate. This contention arises from the absence of substantial therapeutic benefits demonstrated in randomized trials, increased morbidity, and no significant improvements in overall or recurrence-free survival rates [35,36]. These challenges have led clinicians to adopt a less invasive procedure—sentinel lymph node (SLN) mapping. Particularly beneficial for patients with presumed early-stage endometrial cancers, SLN mapping has been shown to offer high sensitivity, specificity, and excellent negative predictive values [37,38,39]. The technique not only reduces morbidity, especially in terms of lymphedema and lymphocele formation compared to complete lymphadenectomy, but also increases the detection rate of nodal metastasis, particularly low-volume metastasis (LVM) [40]. LVM, including isolated tumor cells (ITCs) and micrometastases, represents an approximate 8% increase in nodal positivity over standard pathologic staging and is associated with a better prognosis than macrometastases (>2 mm) [41,42,43].

The FIGO 2023 staging system distinguishes between micrometastasis and macrometastasis, classifying them separately as IIIC1i and IIIC1ii for pelvic lymph node involvement and IIIC2i and IIIC2ii for para-aortic lymph node involvement. Both from a clinical and pathological standpoint, ITCs and micrometastases do not exhibit significant differences in terms of recurrence and survival rates [44]. This justifies the exclusion of ITCs from the new staging system. However, discerning clear benefits of the FIGO 2023 system over the FIGO 2009 system (IIIC1 and IIIC2) remains challenging, particularly since all these cases are currently treated as stage IIIC disease in clinical scenarios.

Many academic institutions in the US have implemented SLN ultrastaging and volume recording of nodal metastases, following the recommendations of the International Society of Gynecologic Pathologists (ISGyP) [45]. Ultrastaging has been found to more sensitively and accurately identify lymphatic disease compared to standard lymphadenectomy [41,46]. Surgical practice is gradually shifting from lymphadenectomy to SLN mapping, an intermediate step before possibly adopting molecular classification for more accurate risk stratification, given the growing evidence of its superiority in providing prognostic data compared to conventional histology [47,48].

It is vital that pathologists continue to efficiently retrieve, process, and identify metastases, including LVM. As per the ISGyP’s recommendations, efficient lymph node retrieval can be accomplished by thorough examination through direct vision, palpation, and sharp dissection. Cutting lymph nodes perpendicular to their long axis at 2 mm intervals increases the likelihood of detecting metastases. Including a rim of adipose tissue surrounding the lymph nodes allows for the evaluation of extracapsular extension when a tumor is present, a feature that could significantly affect prognostic outcomes in FIGO stage IIIC endometrial cancers [49].

## 4. Synchronous Endometrial and Ovarian Endometrioid Carcinomas: Staging Advances and Recent Insights

The co-occurrence of endometrioid carcinomas in the endometrium and one or both ovaries is not an uncommon phenomenon. Differentiating between concurrent independent primary tumors and metastasis, typically from the endometrium to the ovary, relies on various clinicopathological indicators. Traditionally, low-grade tumors have been considered as dual primary carcinomas, associated with a favorable prognosis. Recent molecular studies, however, have suggested that such tumors in the uterine corpus and the ovary are clonal, indicative of metastasis from one site to the other, typically from the endometrium to the ovary [50,51,52]. The implications of this finding are significant, as these clonally related tumors, though likely representative of metastasis from the endometrium, generally have an excellent prognosis. This leads to concerns about potential overtreatment through the unnecessary application of adjuvant therapy, given that they would technically qualify as stage IIIA endometrial carcinomas prior to FIGO 2023.

The paradox presented by these data is puzzling, as the “metastatic” nature of endometrial cancer in these cases deviates from our conventional understanding of metastasis. Furthermore, it is questionable whether evidence of “non-different clones” between cancers in the endometrium and the ovary truly represents metastasis or suggests another unknown mechanism at play. Additional research is necessary to delve into the intricacies of these cases and achieve a deeper understanding of their biological behavior.

In light of this dilemma, the WHO 2020 guidelines [34] advocate for a conservative approach, treating these tumors as synchronous primary neoplasms when four criteria are met: (1) both tumors are low-grade; (2) there is less than 50% myometrial invasion; (3) no other sites are involved; (4) there is no substantial LVSI at any location. The FIGO 2023 staging system incorporated this approach, introducing a new designation, stage IA3. However, it is crucial to remember that the FIGO 2023 staging system explicitly excludes stage IA3 cases with adnexal involvement when conditions such as over 50% myometrial invasion, substantial LVSI, bilateral ovarian involvement, capsular rupture, or the presence of additional metastatic lesions are present. Such cases remain as stage IIIA1, warranting the standard adjuvant treatment.

No staging criteria are infallible, including FIGO 2023. Certain cases can pose challenges in staging, for instance, low-grade EEC cases with 60% myometrial invasion and unilateral ovarian involvement without extrauterine or extraovarian diseases. In these circumstances, the pathologist should communicate the scenario to the oncologist. It is commonplace for such cases to be presented at tumor board conferences, where a collective opinion regarding therapeutic options is discussed. This encourages a comprehensive and collaborative approach to patient management.

## 5. Integrating Molecular Classifications into FIGO 2023 Staging of Endometrial Cancer

The feasibility of integrating The Cancer Genome Atlas (TCGA) molecular classification into clinical practice has been demonstrated through a simplified surrogate testing method [53]. This method employs immunohistochemical markers, namely p53 and MMR proteins, along with molecular sequencing tests to identify pathogenic POLE mutations. The four classifications, POLE*mut*, MMRd, NSMP, and p53*abn*, correlate well with prognostic outcomes of the original TCGA endometrial cancer molecular classification [5,28,53,54,55,56,57,58,59,60]. This approach is pragmatic as p53 and MMR immunohistochemistry can be performed in most pathology laboratories.

In 2021, the European Society of Gynaecological Oncology/European Society for Therapeutic Radiotherapy and Oncology/European Society of Pathology (ESGO/ESTRO/ESP) guidelines incorporated TCGA prognostic groups into the management of endometrial carcinoma [61]. FIGO 2023 fully endorsed these guidelines, advising, though not mandating, the use of molecular classification for all endometrial carcinomas [7]. However, the prerequisite for molecular sequencing to identify pathogenic POLE mutations is still a significant challenge due to many institutions globally lacking this capability. We believe this barrier partially influenced FIGO 2023’s decision to encourage, rather than require, molecular classification for endometrial cancers whenever possible.

This non-mandatory guideline can present dilemmas, as clinicians and pathologists often face uncertainty about when to implement the molecular classification approach. Based on the discussions and experiences garnered at the University of Texas Southwestern Medical Center, there is strong support for the assertion that molecular classification is particularly beneficial and, thus, indicated for the majority of aggressive types of endometrial cancers. These include high-grade EEC, mixed endometrial carcinoma, ambiguous endometrial carcinoma, ECCC, carcinosarcoma, undifferentiated or dedifferentiated carcinoma, and gastrointestinal-type endometrial carcinoma.

Diagnosis of ESC is largely morphology-based, and all ESC cases fall into the p53 abnormality group. Molecular classification also does not significantly benefit patients with MLA, provided the diagnosis is confirmed. For non-aggressive or low-grade EECs with wild-type p53 staining and/or MMR proficiency, POLE mutation testing is unnecessary. This is consistent with the most recently proposed “very low-risk” endometrial cancer (G1/G2 grade, endometrioid type, MMR-proficient, p53 wild-type, stage IA, no LVSI) in whom POLE testing will not impact on patient care [62,63]. However, in the presence of mutant p53 or MMRd staining patterns, performing a POLE mutation analysis is advisable to avoid misclassification into the p53*abn* and MMRd groups [64].

Pathogenic POLE mutation testing is key to this surrogate molecular testing approach. However, not all mutations within the POLE gene are indicative of POLE*mut* endometrial carcinoma. Pathogenic POLE mutations need to meet specific criteria: location within the exonuclease domain and association with an ultrahigh tumor mutation burden (>100 mut Mb). Differentiating between pathogenic and non-pathogenic mutations can be a challenge for many pathologists. Leon-Castrillo et al., from Leiden University, reviewed and summarized all available data, identifying 11 pathogenic somatic missense mutations (Table 1) within the POLE exonuclease domains that classify as POLE*mut* endometrial cancer [55]. Over 95% of POLE*mut* cancers harbor mutations in P286R, S297F, V411L-T/C, S459F, or A456P, with less than 5% showing a mutation in other domains [55,65].

Identifying these 11 pathogenic variants in the POLE gene is critical to determining a positive prognosis and preventing overtreatment in endometrial cancer. Unfortunately, limited access to DNA sequencing methods like Sanger or next-generation sequencing has impeded this effort [66]. Consequently, women with POLE*mut* endometrial cancers often receive overtreatment, leading to treatment-related morbidities and unnecessary costs. This challenge may slow the implementation of the molecular classification and FIGO 2023 in routine clinical practice worldwide [7,67]. To overcome this, Van den Heerik et al. crafted a solution to this issue by developing QPOLE, a POLE hotspot test that is both quick, taking approximately 2 h, and cost-effective. The test uses a quantitative polymerase chain reaction (qPCR) assay [68]. This methodology stands out as an efficient and accurate testing procedure for POLE in endometrial biopsy and hysterectomy specimens. Its global adoptability could aid significantly in making informed decisions about lymphadenectomy and adjuvant therapy. Given its advantages, I am of the belief that the QPOLE assay, when combined with MMR and p53 immunohistochemistry, could effectively substitute POLE sequencings in the molecular classification of endometrial cancers.

In conclusion, the comprehension of endometrial cancer continues to expand with the integration of molecular classifications into the FIGO 2023 staging system. The significant role of histological typing, the subtleties of LVSI, the size of nodal metastasis, and the necessity of distinguishing between synchronous and metastatic cancers highlight the intricacy of this domain. While challenges persist, particularly in regard to the execution of the pathogenic POLE mutation test, advancements like the QPOLE assay are poised to render molecular testing more accessible and clinically actionable. As we progress, the close collaboration between pathologists and clinicians is imperative for the effective interpretation of these advancements, thereby facilitating precision medicine in the management of endometrial cancer and ultimately enhancing patient outcomes.

## Figures and Tables

**Figure 1 cancers-15-04101-f001:**
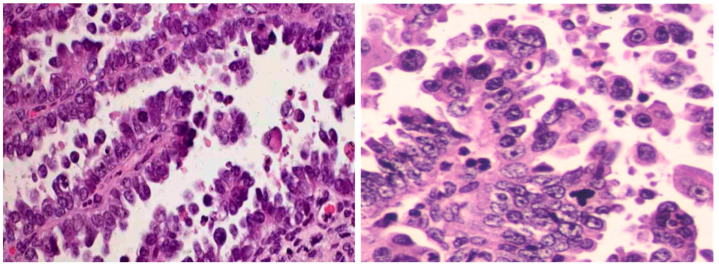
Morphologic Depiction of Endometrial Serous Carcinoma. The two images presented exemplify the classical morphology of endometrial serous carcinoma. On the (**left**), papillary architectures are displayed, lined by high-nuclear grade tumor cells, with some forming micropapillae. The (**right**) panel illustrates an increased number of micropapillae, accompanied by high-grade tumor cells exhibiting macronucleoli. The presence of a few tumor giant cells can also be observed. Molecular classification is generally deemed unnecessary as nearly all such tumors invariably fit into the p53abn category.

**Figure 2 cancers-15-04101-f002:**
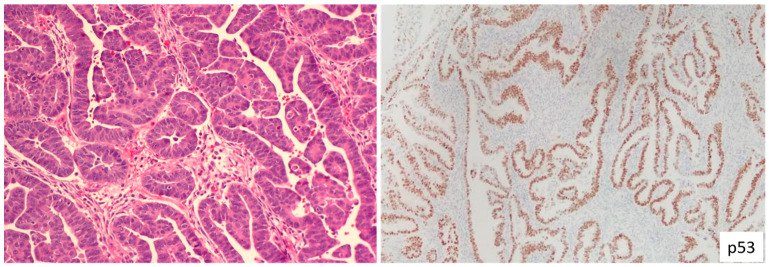
High-Grade Endometrial Carcinoma with Ambiguous Morphology. The (**left**) panel introduces an instance of endometrial carcinoma, predominantly featuring glandular structures lined by high-grade tumor cells. The morphology demonstrates partial characteristics of both serous and endometrioid carcinoma, though it remains indeterminate. The tumor exhibits a mutant p53 staining pattern, as demonstrated by the immunohistochemical evaluation (**right**) panel. It is crucial to note that the observed mutant p53 pattern does not directly equate to a p53*abn* category. Molecular classification of the tumor can only be definitively accomplished through POLE mutation testing.

**Figure 3 cancers-15-04101-f003:**
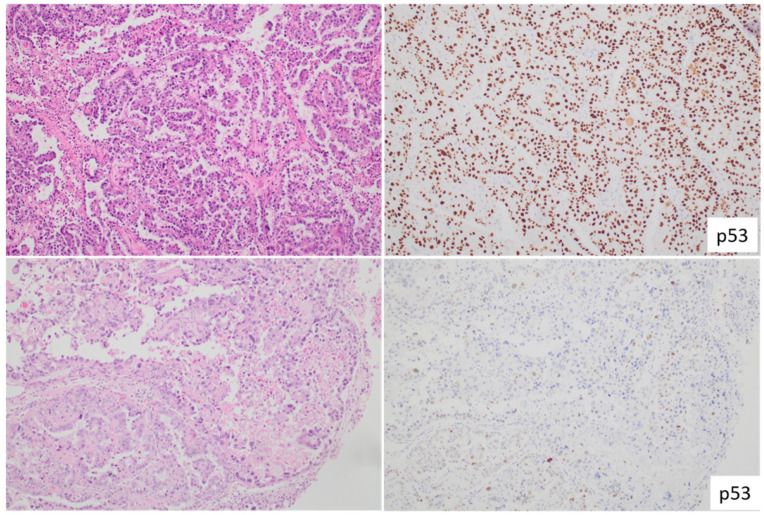
Depiction of Endometrial Clear Cell Carcinoma. The two panels on the (**left**) illustrate two distinct instances of endometrial clear cell carcinoma. One case manifests a mutant p53 staining pattern (**upper right**), while the other case displays a wild-type p53 staining pattern (**lower right**), both identified through immunohistochemistry.

**Figure 4 cancers-15-04101-f004:**
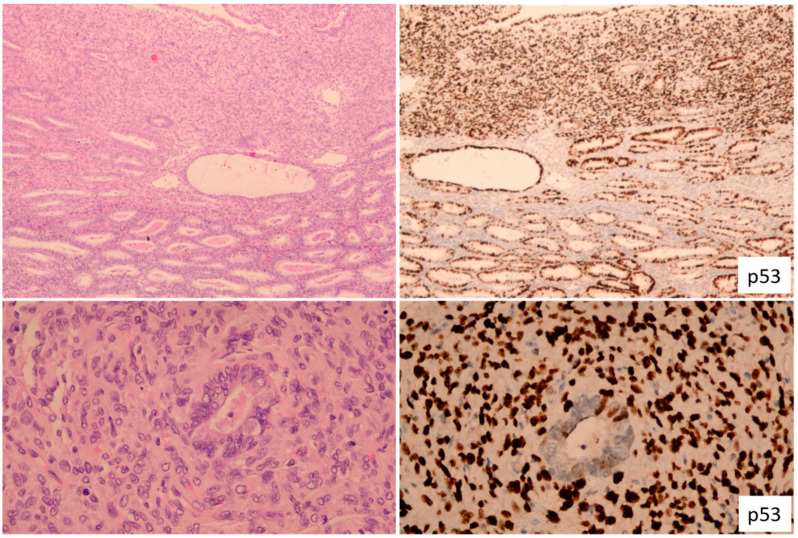
Endometrial Carcinosarcoma. (**Left panels**) exhibit the morphological features of endometrial carcinosarcoma, with the sarcomatous component depicted at the top and the well-differentiated endometrioid carcinomatous component situated at the bottom (as seen in the low-power view, (**upper left**)). A closer inspection reveals a carcinoma gland enveloped within the sarcoma (displayed on the (**lower left**)). The correlating images showcase an overexpressed mutant p53 stain in the sarcomatous component, contrasting with the wild-type p53 present in the carcinomatous component (**right panels**).

**Figure 5 cancers-15-04101-f005:**
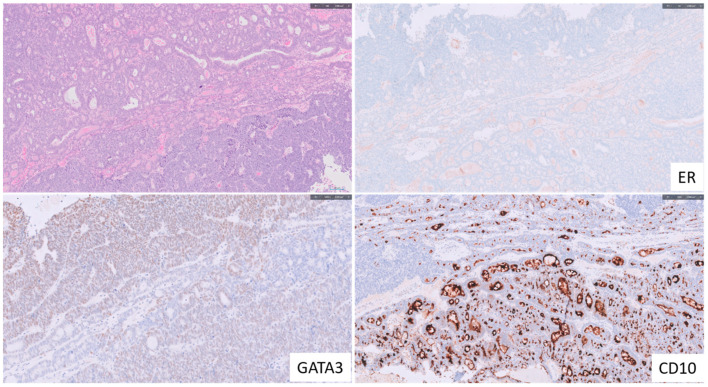
Representation of Endometrial Mesonephric-Like Adenocarcinoma. The (**top left**) panel displays the morphological attributes of endometrial mesonephric-like adenocarcinoma, showcasing glandular tubular formations with a moderate degree of nuclear atypia. The tumor exhibits the typical immunophenotype, with ER negativity displayed in the (**upper right**), GATA3 positivity in the (**lower left**), and CD10 positivity in the (**lower right**).

**Table 1 cancers-15-04101-t001:** Pathogenic POLE Mutations in Endometrial Carcinoma as per TCGA Data, Compiled Following the Study by León-Castillo et al. [55].

Protein Change	No. of Cases	Nucleotide Substitution	Exon	MSI-H Cases (%)	Mutation Recurrence in EC	Mutation Recurrence Pan-Cancer	No. of “Benign” Results by In Silico Tools	POLE Score	EDM	Signature 10 Contribution
P286R	21	c.857C>G	9	1 (4.8)	Recurrent	Recurrent	0	5–6	Y	0.225–0.978
V411L	13	c.1231G>T/C	13	1 (7.7)	Recurrent	Recurrent	1	4–6	Y	0.000–0.751
S297F	3	c.890C>T	9	2 (66.7)	Recurrent	Recurrent	0	5–6	Y	0.123–0.611
S459F	2	c.1376C>T	14	0 (0)	Recurrent	Recurrent	1	5–6	Y	0.940–0.955
A456P	2	c.1366G>C	14	0 (0)	Recurrent	Recurrent	0	5–6	Y	0.277–0.837
F367S	2	c.1100T>C	11	2 (100)	Recurrent	Recurrent	0	6	Y	0.095–0.100
L424I	2	c.1270C>A	13	2 (100)	Recurrent	Recurrent	1	5 or 3	Y	0.000–0.000
M295R	1	c.884T>G	9	1 (100)	Recurrent	Recurrent	0	6	Y	0.785
P436R	1	c.1307C>G	13	0 (0)	Recurrent	Recurrent	0	6	Y	0.230
M444K	1	c.1331T>A	13	0 (0)	Recurrent	Recurrent	0	5	Y	1.000
D368Y	1	c.1102G>T	11	1 (100)	Novel	Recurrent	0	4	Y	0.042

EDM: exonuclease domain mutations; Y = yes; N = no.

## Data Availability

The data presented in this study are available in this article.

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
