# Peer review of "Molecular Classification of Endometrial Cancer and the 2023 FIGO Staging: Exploring the Challenges and Opportunities for Pathologists"

_cancers, 2023, doi:10.3390/cancers15164101_

Round 1

Reviewer 1 Report

Review

Molecular Classification of Endometrial Cancer and the 2023 2 FIGO Staging: Exploring the Challenges and Opportunities for Pathologist

This is an excellent review on molecular classification of endometrial cancer with discussion related to 2023 FIGO staging. The review focus on histopathologic diagnosing of the different types of EC and the importance/relevance of using molecular classification overall and in different subgroups.

This review will be of importance for pathologists working in the field of gynecologic oncology, but also for clinicians responsible for the treatment of women with endometrial cancer.

Author Response

Response:  Thank you for your encouragement.  Appreciated. 

Reviewer 2 Report

This manuscript is very well put together, and I don't see any problems. However, The pathology figures should be more precise.

 I don't see any problems.

Author Response

This manuscript is very well put together, and I don't see any problems. However, The pathology figures should be more precise.

Response:  Thank you for your constructive feedback. I genuinely appreciate your positive remarks on the manuscript. In reference to your comment on the pathology figures, I reuploaded the figures with higher resolution and added them into my manuscript. Please find the attachment.

Reviewer 3 Report

The manuscript : "Molecular Classification of Endometrial Cancer and the 2023 FIGO Staging: Exploring the Challenges and Opportunities for Pathologist" is overall well written and well researched with extensive referencing noted. 

The manuscript provides an excellent summary of the integration of TCGA molecular classification of endometrial cancers with histological subtypes. This includes problems that gynaecology pathologists commonly encounter in the reporting of endometrial carcinomas in biopsy and hysterectomy specimens.

Line 93: "Conversely, NEEC, which includes subtypes such as Endometrial Serous Carcinoma (ESC), Clear Cell Carcinoma (ECCC), carcinosarcoma, gastrointestinal carcinoma, squamous cell carcinoma, dedifferentiated carcinoma, undifferentiated carcinoma, mesonephric-like adenocarcinoma (MLA), and mixed endometrioid with serous or clear cell carcinomas, collectively account for about 20-25% of all endometrial cancers."  Could Neuroendocrine carcinoma be added to the non-endometrioid carcinoma group.  However in relation to the recently published FIGO staging of endometrial cancer 2023 I notice that they also did not include high grade neuroendocrine carcinoma.

Line 154: "Moreover, it's crucial to comprehend that serousoid endometrial cancer, as discerned through molecular classification, does not equate to ESC and includes a significant proportion of p53 mutant EECs [9]. 

This is an important point and emphases the need for the histopathology of the endometrial biopsy to be reviewed by a gynaecology pathologist prior to definitive surgery,  as part of the surgical staging will include an omental biopsy.

It is interesting to read about QPOLE testing which could be a cost-effective substitute for POLE sequencing.

Author Response

Line 93: "Conversely, NEEC, which includes subtypes such as Endometrial Serous Carcinoma (ESC), Clear Cell Carcinoma (ECCC), carcinosarcoma, gastrointestinal carcinoma, squamous cell carcinoma, dedifferentiated carcinoma, undifferentiated carcinoma, mesonephric-like adenocarcinoma (MLA), and mixed endometrioid with serous or clear cell carcinomas, collectively account for about 20-25% of all endometrial cancers."  Could Neuroendocrine carcinoma be added to the non-endometrioid carcinoma group.  However, in relation to the recently published FIGO staging of endometrial cancer 2023 I notice that they also did not include high grade neuroendocrine carcinoma.

Response:  Thank you for the insightful feedback. I acknowledge the significance of endometrial neuroendocrine carcinoma as a unique entity with notably aggressive behavior. Given that the FIGO 2023 system excludes it, I believe it's prudent to remain consistent with that classification in this context.

Line 154: "Moreover, it's crucial to comprehend that serousoid endometrial cancer, as discerned through molecular classification, does not equate to ESC and includes a significant proportion of p53 mutant EECs [9].

This is an important point and emphases the need for the histopathology of the endometrial biopsy to be reviewed by a gynaecology pathologist prior to definitive surgery, as part of the surgical staging will include an omental biopsy.

Response:  I greatly appreciate your highlighting the importance of this point. The nuances in molecular classification underscore the essential role of specialized expertise in gynecological pathology. Ensuring that endometrial biopsy histopathology is reviewed by a dedicated gynecology pathologist not only provides a more accurate diagnosis but also aids in the more informed planning for definitive surgery, taking into account vital steps like omental biopsies. Your emphasis reiterates the importance of a collaborative and specialized approach in the care pathway.

It is interesting to read about QPOLE testing which could be a cost-effective substitute for POLE sequencing.

Response:  I'm pleased to note your interest in the potential of QPOLE testing. I share the optimism that QPOLE, or similar cost-effective molecular alternatives, can pave the way for broader global accessibility, potentially serving as viable replacements for POLE sequencing analyses.

Reviewer 4 Report

A very timely review of an important topic.

The author should include some comments about mixed tumors. How mixed histologies can impact on satiging.  For example, it has ben reported that pure clear cell carcinomas have worse prognosis than mixed endometrioid-clear cell carcinomas, where the frequency of MMRd is much higher.

Author Response

A very timely review of an important topic.

The author should include some comments about mixed tumors. How mixed histologies can impact on satiging.  For example, it has ben reported that pure clear cell carcinomas have worse prognosis than mixed endometrioid-clear cell carcinomas, where the frequency of MMRd is much higher.

Response:  I appreciate the reviewer's insightful comments on the significance of mixed tumors in endometrial cancer staging. It is indeed true that we occasionally come across endometrial cancers presenting with mixed histologies. Among these, mixed endometrioid serous and clear cell carcinomas tend to be more prevalent. I've addressed the matter of mixed endometrioid serous and clear cell carcinomas in Section 1.2 "Histology of Endometrial Cancer: An Independent Prognostic Marker," specifically on lines 96-97. Such mixed histologies, owing to their aggressive nature, are staged based on the FIGO 2023 system.

For further clarity, I've expanded upon this in the "1.4 Endometrial Clear Cell Carcinoma" section, lines 209-210 of the revised manuscript. To quote the updated passage:

"Significantly, all ECCC cases with POLE mutations have shown prognoses similar to other POLEmut endometrial cancers. To prevent potential overestimation of the severity of ECCC including those cancers with mixed histology, molecular classification is recommended whenever feasible. "

Again, thank you for bringing this vital aspect to the forefront, and I trust that these clarifications adequately address the concerns.